# Coming Clean and Avoiding Bubble Trouble–Using Detergents Wisely in the Purification of Membrane Proteins for Cryo-EM Studies

**DOI:** 10.3390/biom15091315

**Published:** 2025-09-12

**Authors:** Bowen Chen, Peter Harrison, Vasileios Kargas, Naomi Pollock, Robert C. Ford, Stephen M. Prince, Richard F. Collins

**Affiliations:** 1Faculty of Biology, Medicine and Health, Smith Building, The University of Manchester, Dover Street, Manchester M13 9PL, UK; 2The Membrane Protein Laboratory and the Electron Bio-Imaging Centre (eBIC), Diamond Light Source, Harwell Science & Innovation Campus, Oxford OX11 0DE, UK; 3Cambridge Institute for Medical Research, University of Cambridge, Cambridge Biomedical Campus, The Keith Peters Building, Hills Road, Cambridge CB2 0XY, UK; 4Health & Life Sciences, Aston University, Aston St, Birmingham B4 7ET, UK; 5Aston Institute for Membrane Excellence, Aston University, Birmingham B4 7ET, UK

**Keywords:** cryo electron microscopy, Cryo-EM grid preparation, membrane proteins, detergent purification, membrane protein structure, Wzz lipopolysaccharide biosynthesis, CorA Magnesium channel structure, transmembrane helices, oligomeric complexes, single particle averaging

## Abstract

Detergent solubilisation remains the most commonly used but potentially problematic method to extract membrane proteins from lipid bilayers for Cryo-EM studies. Although recent advances have introduced excellent alternatives—such as amphipols, nanodiscs and SMALPs—the use of detergents is often necessary for intermediate steps. In this paper, we share our experiences working with detergent-solubilised samples within the modern Cryo-EM structural pipeline from the perspective of an EM specialist. Our aim is to inform novice users about potential challenges they may encounter. Drawing on specific examples from a variety of biological membrane systems, including Magnesium channels, lipopolysaccharide biosynthesis, and the human major facilitator superfamily transporters, we describe how the intrinsic properties of detergent-extracted samples can affect protein purification, Cryo-EM grid preparation (including the formation of vitreous ice) and the reconstitution of proteins into micelles. We also discuss how these unique characteristics can impact different stages of structural analysis and lead to complications in single-particle averaging software analysis. For each case, we present our insights into the underlying causes and suggest possible mitigations or alternative approaches.

## 1. Introduction

Over the past decade, Cryo-EM has become the method of choice for structure/function studies of membrane proteins. While the combination of direct detector cameras [1], improved automated electron microscopes [2], and sophisticated software algorithms [3,4,5,6,7] has made obtaining high-resolution data much easier than in the past, membrane proteins still retain some intrinsic properties that result in practical experimental difficulties. One of the key bottlenecks in studying the structure of membrane proteins is extracting them from their natural lipid bilayer, whilst maintaining their essential native structure and biochemical function [8]. Advances in cryogenic electron tomography (cryo-ET) have made the study of some larger membrane proteins possible in situ [9,10,11]; many projects still require the purification of the isolated protein. Traditionally, detergents have been employed as agents to replace the lipid membrane, facilitating solubilisation and stabilising membrane proteins by acting as a proxy for the membrane environment. Detergents function as amphipathic molecules that interact simultaneously with both hydrophobic and hydrophilic regions of proteins, forming micelles that keep membrane proteins soluble in aqueous solution [12]. Within the body of a micelle, the hydrophobic tails of the detergent molecules partition to form a hydrophobic core, which can then stably integrate hydrophobic compounds such as lipids and the non-polar amino acids within a protein. This central hydrophobic core is shielded from the aqueous environment by hydrophilic head groups on the detergent molecules, which interact with water through hydrogen bonding/electrostatic interactions and stabilise a micelle in solution [13,14]. Micelles are generally spheres or oblate discs, with the akyl chain length essentially determining the micelle dimensions [15]. However detergents can form an array of alternative concentration-dependent structures including inverted micelles, bilayers, and worm-like tubes [16], which are generally not appropriate for single-particle studies.

Prior to the Cryo-EM ‘resolution revolution’ in 2012, the most ubiquitous and successful detergents in membrane protein purifications were β-octyl-glucoside (BOG) and n-Dodecyl-β-D-maltose (DDM) [17]. These are both non-ionic detergents frequently used to solubilise integral membrane proteins, mainly because they are considered ‘mild’; i.e., they pull proteins out of the membrane and maintain protein stability but do not denature them. This mildness is quantified by a combination of biochemical functional assays combined with structural appraisal and contrasts with harsher detergents such as Sodium Dodecyl Sulphate (SDS), which will fully unfold and denature most proteins [18]. There is no absolute correlation between detergent properties, e.g., alkyl chain length, and the propensity for detergents to denature proteins. Detergent properties are assigned empirically based upon their practical utility in the isolation of membrane proteins. Newer detergents have been developed (and have come to dominate usage since ~2015) based on Maltose-neopentyl glycol (MNG) derivatives. Initial studies [19] clearly demonstrated enhanced detergent solubilisation compared with DDM, but the optimal analogue was protein-dependent. Lauryl Maltose Neopentyl Glycol (LMNG) has become the most widely used and successful of these detergents for a variety of reasons [20]. It has preferential physical properties including smaller and more uniform micelles, a flatter profile with dimensions closer to most biological membranes (~45 Å), and a low critical micelle concentration (CMC) which results in fewer free detergent molecules being present, which in turn improves image quality [21]. LMNG also appears to have hydrophobic tails with reduced mobility that enhance the stability of micelles by improving packing around transmembrane helices [22]. More recently, another class of detergent has evolved this base structure again [23], iterating a class of pendant glucose-neopentyl glycols (P-GNG) which solubilise membrane proteins substantially better than DDM and with equivalent characteristics to LMNG. In addition, detergents such as glycodiosgenin (GDN) and digitonin have proven useful in Cryo-EM studies [24]. Detergent use in Cryo-EM structures prior to 2019 was recently examined by a meta-analysis showing that maltosides with and without cholesteryl hemisuccinate predominated, followed by GDN and digitonin and the MNG class [25]. It is also worth noting that micelles of the detergent n-dodecyl-β-melibioside (β-DDMB) may also be useful for smaller membrane proteins; a variety of experiments have shown β-DDMB is ~30 kda smaller than DDM [26]. To address these limitations of detergents, several alternative methodologies have been developed, including bicelles [27], nanodiscs [28,29], styrene maleic acid lipid particles [30], and amphipols [31,32,33]. These innovative approaches offer some distinct strategic advantages in maintaining protein stability and functionality during purification, but as with every research methodology, they do not represent a panacea.

A nanodisc is a description for a nanoscale assembly of phospholipids within a stabilising molecule belt, typically an amphiphilic ‘membrane scaffold protein’ (MSP) [11]. Apolipoprotein-I (ApoA-I) has been widely used as an MSP because it is composed of a repeating series of short amphiphilic helices that can stably wrap around a lipid bilayer, most commonly as two strands organised in an intercalated anti-parallel fashion [34,35]. Nanodiscs are assembled by mixing a detergent-solubilised protein with lipids and an MSP, then removing detergent, in a process akin to reconstitution, into liposomes. Therefore, the preparation of nanodiscs requires an initial use of detergent, can be relatively costly compared to single detergent use, and requires considerable optimisation. Additional purification is usually required after the nanodisc formation to remove empty discs from the preparation. Nanodiscs made with synthetic peptides (Peptidisc) have also been reported [36] as well as saposin A (SapA)-based lipid particles (termed Salipro) [37]. The Salipro system uses the mammalian lipid-binding protein SapA to stabilise and reconstitute membrane proteins (typically helical inner membrane) into a lipid-bilayer-like environment [37,38]. Like MSPs though, these methods also require an initial detergent extraction of the target protein and exchange. Even so, proteins prepared as reconstituted nanodiscs are often stable, functional, and amenable to characterisation by Cryo-EM [29,35].

Styrene maleic acid (SMA) lipid particles (SMALPs) are also composed of membrane proteins embedded in lipid and surrounded by a stabilising amphipathic belt, in this case a styrene-co-maleic acid polymer [30]. SMALPs offer a convenient and cost-effective alternative to detergents for membrane protein solubilization and stabilisation [39,40], though their application in Cryo-EM remains limited. SMALPs form spontaneously when styrene maleic acid copolymers are added to biological membranes, encapsulating membrane proteins and surrounding lipids into ~10 nm discs. As a result, they are sometimes referred to as ‘native nanodiscs’ [41]. Notably, the copolymer is required only during solubilisation; the subsequent buffers are free of both polymer and detergent. Evidence has suggested that SMALPs enhance the structural and functional stability of membrane proteins compared to detergent-based methods, likely due to the retention of native lipids [42,43]. This makes SMALPs a practical tool for membrane protein purification and analysis. Numerous proteins have now been purified using SMA [44,45,46] and related polymers such as di-isobutylene maleic acid (DIBMA) [47,48]. One of the earliest Cryo-EM applications involved AcrB in SMALPs [40,49], followed by the structure of alternative complex III from the respiratory chain of *Flavobacterium johnsoniae*, where protein-bound lipids were clearly resolved in the complex [50]. Despite these successes and their promise, Cryo-EM studies using SMALPs remain relatively rare. Anecdotal reports suggest resolution may be limited, potentially due to the amorphous ice, though this has not been systematically investigated. Some studies have used SMA for solubilisation and purification prior to transferring proteins into amphipols or nanodiscs for Cryo-EM [51]. Additionally, a growing number of alternative polymers are being explored for their potential to improve the results from structural studies. Most of these iterate on the core SMA backbone, but with better control of polymer composition and properties [52,53]. For example, using the diisobutylene maleic acid (DIBMA) backbone generates glyco-DIBMA [54] or native cell membrane nanoparticle (NCMN) polymers [55]. Other variations provide better control of polymer composition and properties with modification to the anhydride groups [55] or may incorporate different monomer subunits, e.g., acrylic acid styrene polymers (AASTYs) and polymethacrylate polymers (PMA) [56]. Several reviews deal extensively with this area and present the advantages and disadvantages of this technology in more detail [57,58].

Amphipols share similarities with both detergents and SMA [59]. They are amphiphilic polymers designed to keep membrane proteins soluble in water when isolated from a membrane, but like detergents they form assemblies without significant lipid content [60]. Rather than forming a micelle to solubilize hydrophobic transmembrane protein regions, they adsorb onto the hydrophobic transmembrane surfaces of proteins, shielding them from water and preventing denaturation [59,61]. No interference from micelles affects the characterisation of proteins prepared in this way, and amphipols, like SMA, are used only once during a purification. However, amphipols rarely directly solubilize proteins from membranes and typically rely on an initial detergent purification. Depending on the protein, amphipols may provide a more stable environment for proteins than detergents, and have been used successfully for structure determination [62,63].

In summary, while working with membrane protein systems, the use of detergents is sometimes unavoidable due to various experimental constraints. When this is the case, it is important to be aware of the potential complications they can introduce, especially in structural studies. This research observation paper highlights several common issues that may arise throughout the Cryo-EM workflow whenever detergents are involved, illustrated in Figure 1. It also offers practical insights, troubleshooting tips, and strategic approaches to help researchers navigate these challenges more effectively and improve the overall quality of their structural data.

## 2. Materials and Methods

### 2.1. Cell Culture, Expression, and Purification of MjCorA

A pET15b expression plasmid encoding *Escherichia coli* codon-optimised full-length MjCorA with a preceding HA tag was prepared by GenScript Biotech (UK) Limited (Unit 9, Kings Meadow, Ferry Hinksey Rd, Oxford OX2 0DP, UK). *Methanocaldococcus jannaschii* CorA (MjCorA) was expressed in *E. coli* cells with the chaperone proteins DnaJ and DnaK and purified using a methodology developed from that described here [64]. Briefly, Competent *E. coli* BL21 STAR cells (Invitrogen™, Leicestershire, UK) were transformed with the chaperone pOFXT7-KJ2 plasmid using procedures recommended by the supplier. BL21 STAR-Chaperone Competent Cells were then prepared by liquid culture and treatment with 0.1 M CaCl_2_. These cells were subsequently transformed with the pET15b-MjCorA plasmid.

Single colonies for MjCorA expression were prepared on agar plates with antibiotic selection. Individual colonies were picked and inoculated into 2× Yeast Tryptone media and grown overnight at 37 °C with shaking. An overnight starter culture was used to inoculate expression cultures then grown at 37 °C with shaking at 200 rpm until the optical density at 600 nm (OD_600_) reached ~0.8. MjCorA expression was induced by adding IPTG (0.5 mM), cultures were incubated for a further 36 h at 20 °C, and cells were harvested by 30 min 4000× *g* centrifugation. Cell pellets were resuspended in lysis buffer (50 mM Tris pH 8.0, 200 mM NaCl, 0.05% 2-mercaptoethanol, 0.5 mg/mL lysozyme, 25 µg/mL DNase I (Roche), 1× EDTA-free protease inhibitor cocktail (Roche cOmplete)) and lysis was performed by probe sonication on ice. The lysate was clarified and membranes were isolated by 1 h ultracentrifugation at 100,000× *g*.

The membrane pellet was resuspended in solubilization buffer (50 mM Tris base pH 8.0, 200 mM NaCl, 0.5% 2-mercaptoethanol, 1% Dodecyl β-D-maltoside (DDM), 1× EDTA-free protease inhibitor cocktail (Roche cOmplete)), ultracentrifuged, and the supernatant containing solubilized membrane proteins was decanted. Affinity chromatography using a gravity Ni-NTA column (Thermo Scientific) was used to purify the solubilized MjCorA protein; the enriched protein was eluted with purification buffer (50 mM Tris base pH 8.0, 200 mM NaCl, 0.04% DDM) containing 0.5 M imidazole.

A PD-10 column pre-packed with Sephadex G-25 resin was used to remove imidazole followed by thrombin digestion to remove the His-tag from MjCorA. The resulting sample was applied directly to a Ni-NTA affinity column and MjCorA protein was collected in the flow-through. Size-exclusion chromatography was performed using a Superdex 200 column on an ÄKTA purifier system (Cytiva) in buffer with 50 mM Tris/HCl pH 8.0, 200 mM NaCl, and 0.15% Dodecyl β-D-maltoside (DDM). Elution was monitored by absorbance at 280 nm, and fractions were analysed by SDS-PAGE to evaluate protein purity. Pooled fractions of MjCorA were concentrated using a 100 kda molecular weight cut-off (MWCO) reconstituted cellulose centrifugal concentrator (Merck, Sigma-aldrich) to give a final MjCorA concentration of 2 mg/mL.

### 2.2. Cell Culture and Protein Purification of WzzB^ST^

Extraction and purification of WzzB^ST^ were performed as previously described with modifications detailed in [65]. The final concentration of the purified WzzB^ST^ protein (20 mM Tris pH 7.5, 150 mM NaCl, and 0.025% dodecyl maltoside (DDM)) was adjusted to give a final concentration of 10 mg/mL.

### 2.3. Sample Preparation for Cryo-EM for MjCorA & WzzB^ST^

Cryo-EM grids were prepared using a ThermoFisher Vitrobit MK IV. Operational details, grids, glow discharging and blotting parameters are all provided in Table 1.

### 2.4. Image Processing and 3D Reconstruction for MjCorA & WzzB^ST^

Three-dimensional structures were produced using RELION 5 [3] using the following general approach, as shown in Table 1: The best image files recorded were imported and CTF-corrected. TOPAZ [66] was used using a model trained on 2000–4000 of the picked particles. Auto-picked coordinates were used to extract particles and initial 2D classifications to clean up the data were performed on samples binned ×4. Preliminary reference-free 3D volumes were generated as start models using the appropriate symmetries and filtered to 60 Å resolution. Heterogeneity sorting of 3D structures was performed and higher resolution structures were calculated on un-binned data once a particle subset was identified. The data were subsequently auto-refined, particle-polished, and post-processed using a data-determined B-factor to produce the final maps. In addition to the gold-standard FSC data produced with RELION, the local resolution of the final map was also alternatively determined by the ResMap algorithm [67], which produced similar global-resolution estimates and effectively provided insight into areas of variability and flexibility with lower resolution. biomolecules-15-01315-t001_Table 1Table 1Cryo-EM and image analysis details for sample preparation and data processing.SampleMjCorAWzzB^ST^**Grid type**C-fFlat T40–200 or 300 Au mesh 1.3/1.2Quantifoil–300 Cu mesh 1.3/1.2 (chloroform- washed)**Glow dDischarge**Emitek K100X: 25 mv for 30 s glow dischargeSolarus II 955: 20 V/5 V with 1:1 H_2_:O_2_ 20 s**Sample vVolume**3 mL**Grid blotting parameters**4–6 s in Vitrobot IV: 5 s wait (100% humidity) at 22 °C2–4 s in Vitrobot IV (95% humidity) at 22 °C**Microscope**KRIOS G2 300 Kv**Sampling**1.06 or 1.072 Å/px1.06 Å/px**Total dose**40 e/Å^2^60 e/Å^2^**Defocus (uM)**−0.75 to –2.25−1.0 to –2.25**Images used**25,000–36,000 total~5000 total**Camera**Gatan K3 or F4i/SelectrisGatan K3**MotionCorr**512 or 1024 FFT box and 5 × 5 patches [68]**CTF correction**CTFFIND 4.1 [69]**CTF resolution**30 Å, 2.5 Å (min, max)**Processing**Topaz for picking [66] & Relion 5.0 with BLUSH [3]**Data bBinning**x4 in process up to 3D refinement**Symmetry**C1, C6, C12C1 and C5C1 & C5

### 2.5. Molecular Modelling and Refinement of WzzB^ST^

Two copies of the full-length WzzB^ST^ atomic models, predicted by AlphaFold2 (AF-Q04866-F1) (PMIDs: 34265844, 34791371), were initially rigid-body-fitted into the Cryo-EM map using ChimeraX (https://www.rbvi.ucsf.edu/chimerax/, accessed on 1 September 2025). While the AlphaFold2 model was fitted well within the density for the first protomer, the second required independent placement of the periplasmic (residues 55–289) and TM (residues 20–54, 289–319) domains followed by real-space refinement of the dimeric WzzB^ST^ model in Phenix [70]. Six copies of the refined homodimer were then fitted into the Cryo-EM map to form a hexameric assembly, which was further refined in Phenix to remove intermolecular clashes and improve the overall geometry. Manual adjustments and visualisation in Coot were also performed before the final Phenix refinement.

### 2.6. Additionally Mentioned Protein Examples—Sample and Image Processing Information

The membrane protein ion channel example (used in Figure 3) was purified into DDM/Tris buffer and concentrated to 2 mg/mL^−1^ before being loaded onto glow-discharged Quantifoil R1.2/1.3 300 mesh Cu grids. The sample was vitrified with a Leica GP2 with a blotting time of 3 s and the grid was screened and imaged with a ThermoScientific Glacios operated at 200 kV with a Falcon 4i detector at 1.192 Å/px.The purified bacterial enzyme example (used in Figure 5) was purified from *E. coli*. For the experiment in Figure 5, the enzyme was solubilised into DDM with the added substrate also solubilised in DDM. A total of 3.5 μL of protein at 1.7 mg/mL^−1^ was applied to a glow-discharged Quantifoil Au 1.2/1.3 300 mesh grid (Harrick Plasma Cleaner, 60 s) and blotted for 2 s before vitrification in liquid ethane using a Vitrobot (Thermo Fisher). EM data were collected on a ThermoScientific Krios operated at 300 kV with a Gatan K3 camera and Bioquantum energy filter. A total of 1604 micrographs were collected at a magnification 130,000× with a physical pixel size of 0.651 Å/px. The total dose for each dataset was 60 e^−^/Å^2^ with a defocus range of −1.0 μm to −2.5 μm and a slit width of 20 eV. Movies were motion-corrected and ctf-corrected using Motion Corr [68] and Ctffind4 as part of the eBIC auto-processing pipeline (PATO) which uses Relion algorithms (https://ebic-pato.diamond.ac.uk/, accessed on 1 September 2025). Particle picking was performed using Blob Picker in CryoSPARC v 3.12 (https://cryosparc.com/, accessed on 1 September 2025), giving 445,289 picks in total. Two-dimensional classification was also performed in CryoSPARC, resulting in a final stack of 35,255 particles.For the experiment shown in Figure 6, the enzyme was purified from *E. coli* and solubilised into LMNG/Tris buffer. A total of 3.5 μL of protein at 2.2 mg/mL^−1^ was applied to a glow-discharged Quantifoil Au 1.2/1.3 300 mesh grid (Harrick Plasma Cleaner, 60 s) and blotted for 2 s before vitrification in liquid ethane using a Vitrobot (Thermo Fisher). Movies were collected on a ThermoScientific Krios operated at 300 kV with a Gatan K3 camera and Bioquantum energy filter. A total of 23,351 movies were collected at a magnification of 130,000× with a physical pixel size of 0.651 Å/px. The total dose was 60 e^−^/Å2 with a defocus range of −1.0 μm to −2.5 μm and a slit width of 20 eV. Movies were motion-corrected and CTF-corrected using Motion Corr 2 and Ctffind 4 as part of the eBIC auto-processing pipeline (PATO) using Relion (as above). A total of 1,748,933 particles were picked in the pipeline and subjected to multiple rounds of 2D classification to yield a final stack of 357,220 particles.The human major facilitator superfamily (MFS) (referred to in Figure 8) transporter was purified from the Baculovirus expression system. Protein was concentrated to 5.1 mg/mL^−1^ and grids were prepared using an SPT Labtech Chameleon using Quantifoil Active 1.2/0.8 grids, glow-discharged internally in the Chameleon. EM data were collected on a ThermoScientific Glacios operated at 200 kV with a Falcon 4 detector at 1.192 Å/px. A total of 1059 movies were collected at a magnification of 130,000× corresponding to a physical pixel size of 0.651 Å/px. The total dose for each dataset was 60 e^−^/A^2^ with a defocus range of −1.0 μm to −2.5 μm and a slit width of 20 eV. Movies were motion-corrected and CTF-corrected using Motion Corr 2 and Ctffind 4 as part of the eBIC auto-processing pipeline using Relion (as above). A total of 530,402 particles were picked and subjected to 2D classification.

## 3. Results and Discussion

Problems with membrane protein samples often only become apparent once a project moves to EM with grid preparation and image acquisition. At this stage, issues such as the dominance of empty detergent micelles in vitreous ice (**Issue 1**) or particle behaviour on grids (**Issue 2**) may emerge in the EM data. At the University of Manchester, the Cryo-EM platform works closely with the BIOMOL platform to support researchers through this critical early phase: an approach we recommend as best practice (see [71]). To minimise these challenges before microscopy begins, a comprehensive buffer and detergent screen (e.g., Vitro-ease has a six-detergent matrix) can help identify optimal conditions (cf. [72]). In our experience, around 50% of new projects start with sub-optimal buffer or detergent combinations which can be readily improved. Early screening is essential before investing in Cryo-EM, where consumables (e.g., grids, C-clips, auto-grid cartridges) can cost approximately GBP 400 per 10-grid batch. When combined with Mass Photometry (e.g., REFYN [73]), this screening can provide valuable insights into sample degradation, aggregation, oligomeric state, and purity, saving significant staff time and avoiding unproductive efforts.

### 3.1. Issue 1: Empty Micelle ‘Up-Concentration’

A related challenge to grid preparation issues arises when samples become overwhelmed by excess empty detergent micelles—despite purity metrics suggesting otherwise. In Figure 2, we illustrate a clear example we encountered during the initial purification of the divalent metal ion transporter protein MjCorA in DDM. Panels 2A–C clearly show the protein is present at high purity and at concentrations that should be readily visible by Cryo-EM. The purification followed a multi-step protocol previously optimised for sample stability and solubility [64]:His-tag purification on a Ni^2+^ affinity column;Tag cleavage and removal;A final size-exclusion chromatography (SEC) step to enhance purity and homogeneity but also reduce the DDM concentration from 0.04% to 0.015%.

**Figure 2 biomolecules-15-01315-f002:**
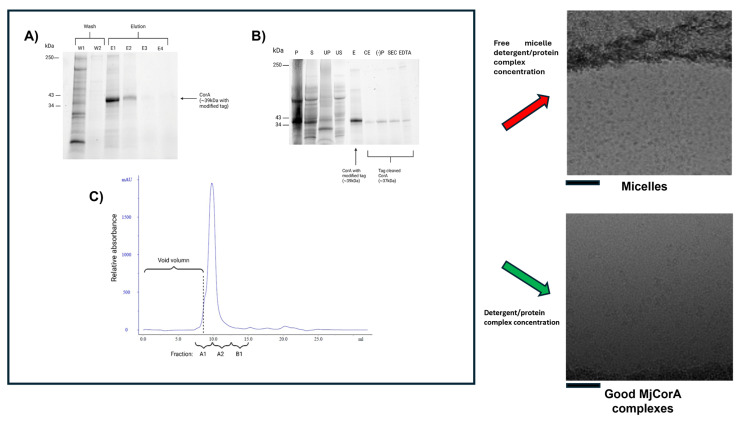
Micelle domination at the end of purification. (**A**) Purification of MjCorA protein via Ni-NTA affinity. SDS–polyacrylamide gel electrophoresis (SDS-PAGE) of fractions from the affinity purification of the MjCorA protein are shown. W1/2: Wash steps; E1-4: elution with imidazole. (**B**) SDS–polyacrylamide gel electrophoresis (SDS-PAGE) illustrating the purification of the MjCorA protein. Lanes show representative fractions from each purification step. P: Pellet of *E. coli* lysate; S: supernatant of lysate; UP: membrane isolation ultracentrifugation pellet; US: corresponding ultracentrifugation supernatant; E: pooled eluant from NiNTA affinity; CE: product after thrombin digestion; (−)P: flow-through of negative purification; SEC: pooled eluants after size-exclusion chromatography; EDTA: EDTA-treated MjCorA. (**C**) Size-exclusion chromatography analysis of MjCorA. Fractions analysed by SDS_PAGE for purity are labelled; those pooled for Cryo-EM analysis are A1 and A2. Panels to the right show the effect of Centricon treatment on the final Cryo-EM data fields. Image scale bars = 500 Å. Raw gel data is provided in Appendix A.

Detergent was maintained throughout to preserve membrane protein solubility. However, early Cryo-EM grid screenings revealed a consistent result with no visible protein. The EM field was dominated by uniform, empty micelles (Figure 2, top-right panel). Initially, we used excess detergent in chromatography buffers to ensure the protein remained soluble and to prevent aggregation. After affinity purification, detergent levels were reduced to approach the critical micelle concentration (CMC) during SEC. Since DDM has negligible absorbance at 280 nm and is displaced by an excess of SDS in the preparation of proteins for SDS–polyacrylamide gel electrophoresis, these micelles remained essentially invisible during purification.

Systematic control experiments eventually linked the issue to the final concentration step using centrifugal concentrators: Centricons (or equivalent membrane concentrators). In our hands the important factors are the use of a reconstituted cellulose membrane with a large molecular weight cut-off. Free micelles and monomeric detergent pass through the concentrator membrane more readily than the protein detergent complex, thus preserving the solution conditions for the membrane protein whilst enriching the detergent protein complex in comparison to free micelles. The size and aggregation number of a DDM micelle has been determined by neutron scattering [74] giving an approximate micelle molecular weight of 66 kda. Using 100 kda concentrators pre-washed with detergent-free buffer before use reproducibly eliminated the problem of excess free micelles (Figure 2, lower-right panel). This subtle but significant artefact is one we have encountered across multiple membrane protein systems (P-gp, CFTR, WzzB^ST^, MjCorA). We recommend that novice users consider this possibility if they observe similar issues—especially when their Cryo-EM grids appear dominated by micelles rather than being populated with an identifiable protein.

### 3.2. Issue 2: ‘Concave Lensing’ of Vitreous Ice in the Preparation of Cryo Grids

Preparing vitreous ice on cryo-specific holey EM grids such as Quantifoils or C-flats [75,76] can produce a sample spreading problem with uneven particle distribution that is endemic in cryo studies of membrane proteins (Figure 3A,B). When a detergent is added to a buffer, one physical effect is the lowering of the surface tension of the solution, as the hydrophobic tails of the detergent molecules partition to the air–water interface [77]. This has the effect of reducing the surface tension of a solution, such that it will spread much easier [78]. On Cryo-EM grids, we frequently observe a phenomenon we term ‘*concave lensing*’, shown in Figure 3.

In this membrane protein sample, the decreased surface tension of the solution across the Quantifoil hole has caused the solution to thin from the centre outwards forming a concave lens shape across the hole. This is probably exacerbated by the slightly bevelled edge apparent in Quantifoil grids. This has several sample effects that can be observed in the data. Firstly, the centre of the hole has very thin ice that gradually thickens outwards as the vitreous ice is tracked towards the carbon surface. From a cryo processing point-of-view, this thin amorphous ice is problematic because it is usually unstable and will burn or rip in an electron beam very easily [68,79]. If it does not burn or rip, it will usually produce excessive particle motion in the vitreous ice which may not be correctable [80]. Secondly, the central volume within the thin amorphous ice section is so small the protein may not physically fit in it. The amorphous ice then pushes the sample towards the edges of the carbon in a gradient and this has the effect of overlapping the particles in layers. Finally, this pushing effect in a limited gradient can exacerbate problems with particle orientation and Figure 3 shows the particles are nearly exclusively orientated in a top-view orientation which makes 3D processing impossible. The particles around the lip of the thinnest central area have congregated around with a regular interparticle distance indicating a boundary interaction. Other studies have shown an optimal concentration of detergent between 0.05 and 0.4% _(*w*/*v*)_ and that the presence of a low concentration of detergent with a high critical micellar concentration may protect proteins from denaturation effects at the air–water interface [81]. We have found this problem to be frustratingly intractable and have on occasion used continuous carbon grids as alternatives. These grids are usually modified Quantifoils or c-flat-type grids with a very thin continuous layer of carbon or graphene/graphene oxide over the hole. The sample is then placed on the carbon with a shorter blot time. In our experience, this approach has not usually improved grid quality and is particularly prone to exacerbating issues with preferred orientation.

The most commonly used instrument for preparing Cryo-EM grids is the Vitrobot [82]. It is worth highlighting a detail often overlooked by inexperienced users: visually inspecting the blotting papers used during excess sample removal (Figure 3C). The Vitrobot MK IV controls amorphous ice thickness using a simple method and robotically presses two discs of filter paper onto a Cryo-EM grid suspended by tweezers within a humidity control chamber. The blotting is thus controlled by the length of blot time and how tightly the two discs of filter paper come together (the blot ‘force’). With proteins in detergent solution, it is common to observe that standard blotting conditions produce larger overlapping blot circles, as the sample tends to spread more readily, as shown in Figure 3C. In our experience, reducing the blot force slightly (thereby applying less pressure to the grids) can help mitigate this effect and in turn reduce the lensing artefact on imaged grids. When these problems inevitably arise, we have found that a systematic, matrix-style approach—testing variables such as grid type, grid-bar metal, hole size, and spacing—can be particularly effective. Alternative Cryo-EM grids such as UltrAuFoil and HexAuFoil [75,83] feature full metal surfaces and support smaller hole diameters of 0.6 µm and 0.31 µm, respectively, offering promising alternatives. The hexagonal packing of HexAuFoil and reduced hole size may promote more uniform amorphous ice distribution within the holes by reducing directional stresses within the vitreous ice. This can be especially advantageous for membrane proteins and may help prevent the vitreous ice buckling under electron beam irradiation.

Another issue that may arise, even where concave lensing is avoided, is ‘*interface accumulation*’. This phenomenon can occur with all proteins but is more commonly observed in detergent-based membrane protein preparations. It typically presents as a ring of protein adhering to the carbon/ice boundary, as shown in Figure 4. The underlying causes of this phenomenon remain unclear. However, there is a notable charge difference between glow-discharged carbon and the thin buffer layer and this may provide a preferential environment for many proteins. While not as experimentally limiting as lensing, interface accumulation can still significantly reduce the number of ‘pickable’ particles available for analysis, because so many are pushing against the rim of carbon. As a result, more images and additional initial processing time may be required. It is also worth considering using a different grid preparation system in these cases. Most labs use the TF Vitrobot as their work-horse instrument for grid preparation because they are relatively cheap and usually produce usable grids to bootstrap into a project [82]. Some other possibilities include the Chameleon [84], Cryogenium [85], or Spotiton [86], which may be particularly advantageous for detergent membrane protein samples. The Chameleon robot uses blot-free ‘self-wicking’ grids and an nl scale direct volume dispenser onto the grid surface, so excessive spreading of samples can be avoided. The wicking generally occurs on the millisecond scale compared to seconds for the Vitrobot and so ice thickness can be more precisely monitored in real time. However the Chameleon does generally require a higher protein concentration (>5 mg/mL) and so up-concentration of micelles can emerge as a new problem (see Issue 2). The Spotiton system uses a completely different approach and uses an inkjet-style spray of the sample in the picolitre-to-nanolitre-volume scale and directly applies the sample to the grid in such small amounts blotting is not required [86]. Excessive sample spreading and the formation of ‘concave lensing’ remains the most challenging issue for membrane protein grid preparation. If these alternatives fail to resolve the issue, switching to a different detergent or solubilising agent may be the only viable solution.

### 3.3. Issue 3: Empty Micelle ‘Swamping’ of 2D Classification and Unusual Micelle Structures

As discussed in the previous section, when handling detergent-solubilised membrane proteins, the detergent must be kept at a concentration above the CMC to stabilise the bulk solvent behaviour of the sample. The free, empty micelles, when concentrated, are also observable in micrographs. For some membrane proteins, especially those which are small or completely embedded within the membrane with little-to-no protruding protein densities, these empty micelles can often be visually indistinguishable from micelles containing proteins. In these systems, the full extent of the issue is often only apparent at the 2D classification stage of data processing, where secondary structure features like transmembrane helices in the micelle should become readily visible.

Figure 5 highlights this issue in a small-membrane protein. Even after running through two rounds of 2D classifications and reducing the dataset size by ~93%, the sample remains contaminated with empty micelles, essentially because the shell of the micelle head layer is a strong low-resolution feature with high contrast. In ‘normal’ datasets, repeated rounds of 2D classification can be used to clean particle datasets from contaminating particles, aggregates, or carbon rim, but this is computationally expensive and time consuming. The domination of holes with empty detergent micelles reduces the number of particles of protein in the image, requiring, in effect, more data to be collected. This issue can become more pronounced when concentrating low-yielding proteins after SEC, or when adding detergent-solubilised hydrophobic substrates to a membrane protein sample. An attempt to improve the sample in Figure 5 was made through purification in DDM with the incorporation of a lipidic substrate in an attempt to obtain a co-structure. However, following the final purification steps, a high concentration of detergent micelle was still present in the solution. Only after repeated rounds of 2D classification was it possible to distinguish the protein from the empty micelles. If this problem persists, it is worth swapping between different software packages and trying different variations on the 2D class sorting.

Figure 6 presents an example where unusual detergent micelle architectures can be observed in the micrograph images. In this example, the detergent LMNG has formed ‘traditional’ structures which are to be expected, as highlighted in the red box, but additional larger structures have also become apparent during 2D classification. Non-ionic detergents like LMNG are known to be able to form elongated, worm-like structures [81,87]. The appearance of these structures is related to the over-concentration of the detergent used (which can increase during a final concentration step, as noted above). Although these larger structures are usually clearly visible in the micrographs and may not appear to cause an issue initially, they can cause issues during particle classification and add more time to the data processing pipeline. Careful optimisation and possible lowering of the detergent concentration used should alleviate any issue.

In general, removing excess detergent always presents a challenge, since the detergent must be present above the CMC to prevent protein denaturation. As such, one must always have an excess of detergent present and free micelles will almost always be present—the goal is to minimise this. One possible solution to this issue is to employ ion-exchange chromatography as an additional step following gel filtration chromatography. The protein can be bound to a column and eluted in a smaller volume, thus concentrating the protein without concentrating detergent micelles. Additionally, the concentration of detergent can be lowered at the gel filtration step to be closer to the CMC (for example, ~1.5 × CMC) [88]. Similarly, gradient centrifugation approaches such as GraDeR can be used to separate empty detergent micelles from detergent-solubilised proteins [89]. The suitability of these approaches is likely to be protein-dependent. Additionally, low-CMC detergents can also be trialled. Lauryl Maltose Neopentyl Glycol (LMNG), for example, is often used for membrane proteins as it has a low CMC and is difficult to displace with another detergent. As such, the detergent remains bound to the protein even in the absence of excess detergents. Performing the final gel filtration step in buffers without detergent can be considered; however this is likely to result in sample aggregation and denaturation. If a researcher does persist with this type of data, it may be worth trying to change the sensitivity of the resolution and CTF parameters to bypass the very-low-resolution signal micelles can contribute in the image processing software, but this can create as many problems as it solves.

### 3.4. Issue 4: Projection Contamination from Detergent-Driven Oligomerisation

Membrane proteins often exhibit unexpected associations, including the occasional formation of pseudo-D-symmetrical arrangements. D symmetry is a combination of two separate rotational symmetries; the first is around a central long axis (usually vertical), where the complex has Cn rotational symmetry and then a second additional two-fold rotational axis perpendicular to the main axis. D-symmetry is frequently found in modular enzymes [90], and in contrast to complexes like chaperones where the D-symmetry is related to structure and allosteric control (e.g., GroEL), in membrane proteins, it is likely artefactual and non-physiological. Although in highly curved membrane structures found in mitochondria or chloroplasts, the folding means it may be possible for proteins to naturally form these contacts.

Figure 7 presents two distinct examples of such non-physiological interactions, each demonstrating different structural interfaces: (A) The capsular polysaccharide protein complex WzzB^ST^ forms an end-to-end assembly via the micelle-facing surface; (B) the divalent ion channel MjCorA exhibits a similar end-to-end interaction, but through its cytoplasmic extensions, or “legs,” which protrude from the membrane. From a 2D image processing standpoint, these interactions can complicate particle averaging. ‘Contaminated’ side-views can be discarded or tightly masked, but top-views can persist. The interface is also not exact, so there is some slide resulting in less precise alignment. It is apparent as the projections start to tilt that the contaminating underlayer influences signal, shown in Figure 7A (lower panel). In strictly side-on orientations, particles can be selected, aggressively masked, and centred to exclude density contributions from adjacent particles. However, challenges arise when side-views are partially tilted, making it difficult to distinguish overlapping densities. This issue is exacerbated in top-views looking down the long axis, where contaminating density from underlying particles is harder to eliminate. In the case of WzzB^ST^, the pseudo-D-symmetry is approximate and slides, reducing the resolution in both halves of the pseudo complex, necessitating more aggressive data pruning. This reduces the achievable resolution or requires larger datasets to compensate. For the MjCorA, some interactions appear more specific, with discernible secondary structures at the interface between complexes, meaning the D-symmetry has precise interaction but cannot be physiological because of the arrangement of the cell membrane in vivo. These top-view particles cannot be reliably used and should either be excluded from the dataset or subjected to more complex particle subtraction workflows. Both these scenarios increase computational demands, particularly GPU usage and processing time.

A surface-rendered reconstruction of a WzzB^ST^ complex, generated using loose particle selection and without masking, illustrates how these artefacts manifest in 3D processing (Figure 7A, right). Compared to the final high-resolution dataset (see Figure 10 later for the equivalent structure), the resolution is lower, and a diffuse density is visible beneath the micelle—attributed to contributions from misaligned, overlapping particles.

### 3.5. Issue 5: Irregular Micelle Insertions

Another 2D processing issue that only becomes apparent after substantial analysis is that protein insertion into micelles is not always uniform or predictable. As shown in Figure 8, proteins can exhibit unusual organisations within micelles. For example, in Figure 8A, the protein sits asymmetrically, off-centre within the micelle. Micelles are flexible, dynamic structures that can deform easily or even merge with one another, so this behaviour can be expected. It can, however, lead to uneven detergent distribution, especially around proteins with irregular or tilted transmembrane helices (unpublished observations on ABCB6) or amphipathic surface regions. In some cases, particularly when using a high protein-to-detergent ratio, micelles may encapsulate multiple protein complexes (Figure 8B).

Additionally, larger micelles can accommodate proteins in random up/down orientations (Figure 8C), further complicating analysis. Unlike micelles, native membranes have intrinsic lipid asymmetry. For instance, the *E. coli* outer membrane contains lipopolysaccharide (LPS) in the outer leaflet and phospholipids in the inner leaflet. This asymmetry is essential for membrane function and is actively maintained by dedicated lipid transport proteins [91]. Interestingly, the first high-resolution structure of human CFTR was only achieved after adding 0.2% cholesterol to the detergent LMNG [92]. Cholesterol stabilises membranes by intercalating between lipid tails and broadening the membrane’s phase transition temperature [93]. In contrast, micelles are chemically uniform and lack the structural complexity of a lipid bilayer.

These micelle-related issues can lead to significant processing inefficiencies and wasted computational effort.

### 3.6. Issue 6: Variable Micelle Dimensions in 3D Structures

During 3D heterogeneity classification in software such as RELION or CryoSPARC (discussed in [94]), the primary goal is to sort datasets into distinct conformational states, separate intact complexes from partially assembled or damaged ones, and enhance resolution by focusing on more homogeneous subsets of data. While achieving the highest possible resolution is often the main objective, the discarded classes can also provide valuable insights into sample characteristics.

This becomes especially relevant when working with detergent-solubilised proteins, where micelle variability often dominates these analyses. Figure 9 illustrates this using an initial 3D classification of an MjCorA dataset. Three preliminary structures were generated from a cleaned 2D dataset of ~550,000 particles. At a lower resolution, the main difference among them is the width of the DDM micelle. In this case, the pentameric protein complex is relatively rigid and conformationally stable, with very little difference, while the micelle varies in size by ~15% in the X/Y plane. Although later processing steps—such as selective masking, particle subtraction, and BLUSH [3]—can reduce these effects, early analyses are often confounded by micelle variability, which can obscure meaningful protein differences and complicate interpretation.

### 3.7. Issue 7: Unusual Symmetries

The common assumption when extracting a membrane protein into detergent is that a single functional unit—monomeric or oligomeric—is reconstituted into one micelle. However, as shown in Figure 8B,C, this is not always the case. A further complication is the ability to generate high-resolution structures whose biological relevance remains unclear (Figure 10).

WzzB^ST^ is a bacterial inner membrane protein involved in lipopolysaccharide (LPS) O-antigen biosynthesis, a key component of the Gram-negative outer membrane. As a polysaccharide copolymerase, WzzB^ST^ regulates the modal chain length of O-antigen polysaccharides, which are critical for virulence, immune evasion, and resistance to environmental stress [95]. It does not polymerize sugars directly but ensures consistent chain length. Previous Cryo-EM studies ofWzzB^ST^and its homologs have revealed oligomeric assemblies forming either C4 octamers [96], C8 octamers [97], or C12 dodecamers [65]. The physiological relevance of these forms remains unclear, though the C8 symmetry aligns with other LPS assembly components such as Wza [98] and the most recent structure reveals that the protein WzyE, a glycosyltransferase, sits within the chamber at the periplasmic face and controls polymerisation of polymers presumably extruded through Wzz like a nozzle to the outer membrane components [96]. In our earlier C12 dodecameric dataset [65], additional analysis has revealed the sub-organisation of a dodecamer with C6 symmetry, with the dodecamer forming a ring of dimers within the micelle. Our previous work reported the WzzB^ST^dodecamer at 9 Å resolution. In this follow-up, we improved resolution to ~4.2 Å, primarily through better sample preparation: cleaner grids, higher particle concentration, and thinner ice (see Figure 4). The improved algorithms in BLUSH also allow for better particle separation on other projects [99] and on occasion excessive particle binning in early analysis can overlook subtlety in particle orientations and organisation.

This improvement was evident immediately in the reference-free 2D classes. Side-views revealed helices in both periplasmic and transmembrane regions, while top-views showed 12 radially arranged subunits that displayed a distinct hexagonal shape, indicating C6 symmetry despite the dodecameric composition (Figure 10A). There was another smaller oligomer present in the sample, shown in Figure 7A. We were unable to solve this due to the lack of complementary orientations but the size estimates were consistent with a tetramer. The 3D structure of the WzzB^ST^ dodecamer revealed a bell-shaped hollow chamber oriented toward the periplasmic face of the inner membrane, consistent with previous structural work. While earlier work applied C12 symmetry [65], the higher-resolution data of this subset now shows that the oligomer in this preparation forms a ring of six dimers (Figure 10B). The two transmembrane helices form a unique cross-braced motif, further stabilised by opposing cytosolic N- and C-terminal amphipathic helices. Oligomerisation is primarily driven by periplasmic domain residues, with each dimer associating via a large interface dominated by α-helical elements. Notably, no subunit–subunit contacts are observed within the lipid-embedded region. The overall dodecameric complex is offset into a corrugated C6 ring of WzzB^ST^dimers, though this symmetry is largely restricted to the periplasmic region; the transmembrane domains and termini show only minor deviations from C12 symmetry. Figure 11 illustrates this phenomena with volume slices through the height of complex.

The transmembrane helices appear aligned not only by adjacent periplasmic domains but also by the cytoplasmic N- and C-termini, which stabilise the structure on the cytoplasmic face. These features are visible from the cytoplasmic view, where clear densities sit beneath the detergent micelle, perpendicular to the C6 symmetry axis (Figure 10B and Appendix A). The atomic model reveals a well-defined amphipathic helix at the N-terminus and a more loosely structured, marginally amphipathic helix at the C-terminus (Appendix A), which correspond to these well.

In summary, the structure, while interesting, is something of an enigma. It is novel, distinct from previously published forms, and marked by a unique C6 arrangement. This hints that the complex may form monomers or dimers capable of supporting a ratcheting assembly mechanism in vivo, or has several conformational variants that may assemble or disassemble throughout the different stages of LPS synthesis. While dimer contacts are clearly driven by protein interactions outside the micelle, the micelle itself adopts an unusual donut-like micelle torus, in contrast to the typical disc-like detergent micelles. The detergent may allow the protein to adopt different symmetries within the same oligomeric form resulting in the effects seen in Cryo-EM at high resolution. Clearly this system needs further refinement and highlights the ambiguity that using detergent can bring to a system. An alternative solubilisation system would help in this case to resolve the significance.

## 4. Conclusions

Detergents will remain essential in membrane protein purification for the foreseeable future. Their simplicity makes them the preferred starting point in many Cryo-EM studies and even with alternative methodologies (except for SMALP), detergents are still required at the initial extraction stage of purification. This means the specific challenges highlighted in this paper, including surface tension, micelle ‘passengers’, and processing artefacts, will continue to be relevant.

Future advances in detergent chemistry may come from approaches that focus on better replicating the complexity of the native membrane’s hydrophobic core. One promising development avenue is the emergence of tandem triazine maltoside (TZM) detergents, which densely pack within the alkyl chain and can stabilise traditionally challenging proteins [100]. Variants of these TZMs featuring a third hydrophobic alkyl tail and a tunable spacer group show similar potential [101]. Another novel class of detergents, 3,4-bis(hydroxymethyl)hexane-1,6-diol-based maltosides (HDMs), have been developed and initially tested on G-protein-coupled receptors (GPCRs). HDMs were superior in membrane protein extraction to DDM, but were comparable in performance to LMNG [102]. It is also possible hybrid systems combining detergents with polymers (e.g., amphipols) may emerge which preserve native-like environments while maintaining solubility.

In this review, we have examined the challenges associated with the use of detergents in cryo-electron microscopy (Cryo-EM) and discussed practical strategies to address them. Enhancing the efficiency of membrane protein structural biology requires both a clear understanding of these obstacles and the implementation of targeted mitigation approaches, such as sample optimisation informed by biophysical analyses and the development of novel technologies. These efforts are essential for addressing the fundamental biological questions that membrane proteins are uniquely involved in.

## Figures and Tables

**Figure 1 biomolecules-15-01315-f001:**
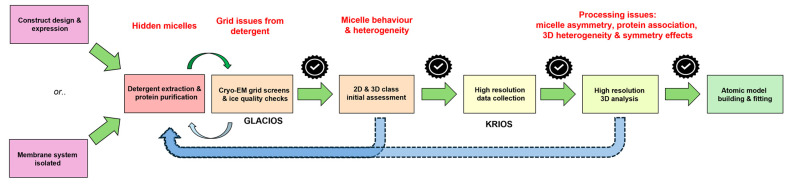
Process flow through the Cryo-EM pipeline for a detergent-solubilised membrane protein. The flow diagram illustrates the points in the process where problems directly attributable to detergent usage can manifest themselves (red text). The process is iterative with sample feedback informing changes required to improve sample quality at distinct stages in the pipeline.

**Figure 3 biomolecules-15-01315-f003:**
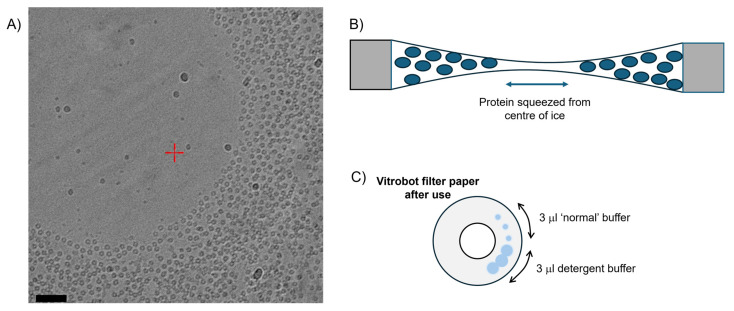
(**A**) Micrograph of a detergent-solubilised membrane protein in a hole on a Quantifoil 1.3/1.2 carbon grid demonstrating ‘concave lensing’. Image courtesy of Dr. Jessica Kleiz-Ferreira (KU Leuven), with imaging conditions described in the Methods. Scale bar = 500 Å. (**B**) The thin empty ice in the middle of the hole forces the protein (an invertebrate pentameric ligand-gated ion channel) to the edge of the hole, where it has adopted a preferred orientation, shown in the cartoon. (**C**) Blotting patterns on used Vitrobot Mark IV filter paper—the spreading tendency of detergent solutions can be seen as larger overlapping blotting spots.

**Figure 4 biomolecules-15-01315-f004:**
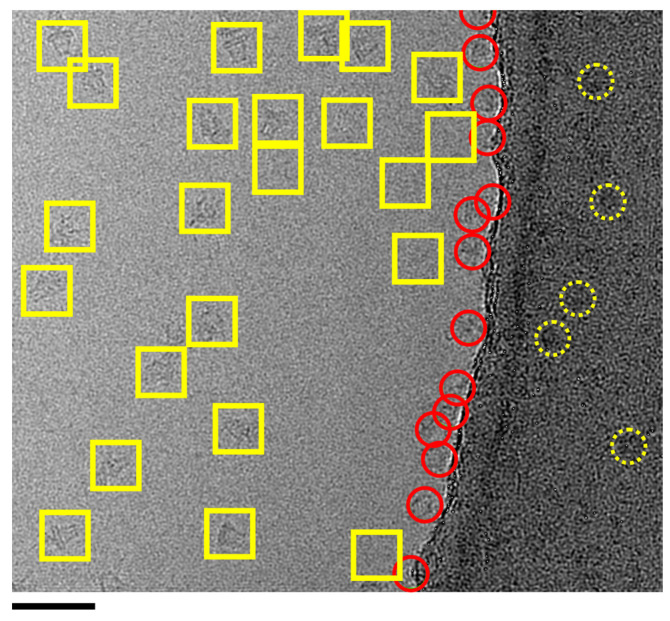
Particle accumulation along the carbon rim at the ice/carbon interface. An example EM field of WzzB^ST^ has avoided concave lensing, but ~40–50% of observable particles have accumulated around the edge of the carbon (red circles). Complexes picked for analysis are shown in 200 Å yellow selection boxes. Particles can also be observed on the carbon—some examples are shown in dotted circles. Scale bar = 500 Å.

**Figure 5 biomolecules-15-01315-f005:**
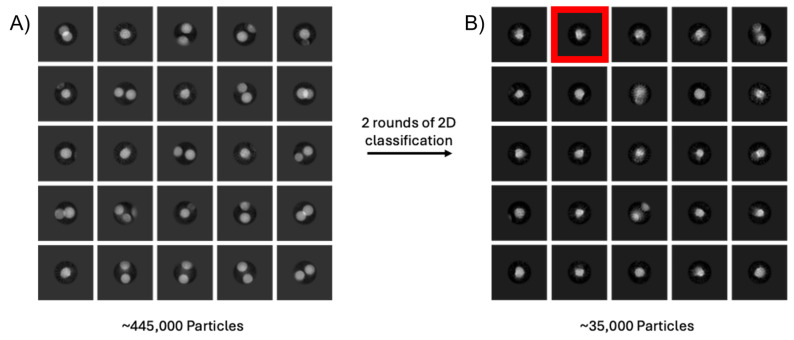
Micelles may ‘swamp’ out repeated 2D classifications of a bacterial-membrane-bound glycosyltransferase enzyme. (**A**) Results of the first round of 2D classification are shown on the left (subset of the entire dataset) where only micelles are apparently visible due to the averaging of the 2D classification. On the right (**B**), following repeated 2D rounds of classification, some protein classes are visible (highlighted in red box), but micelles still dominate the classes. Box = 462 Å^2^.

**Figure 6 biomolecules-15-01315-f006:**
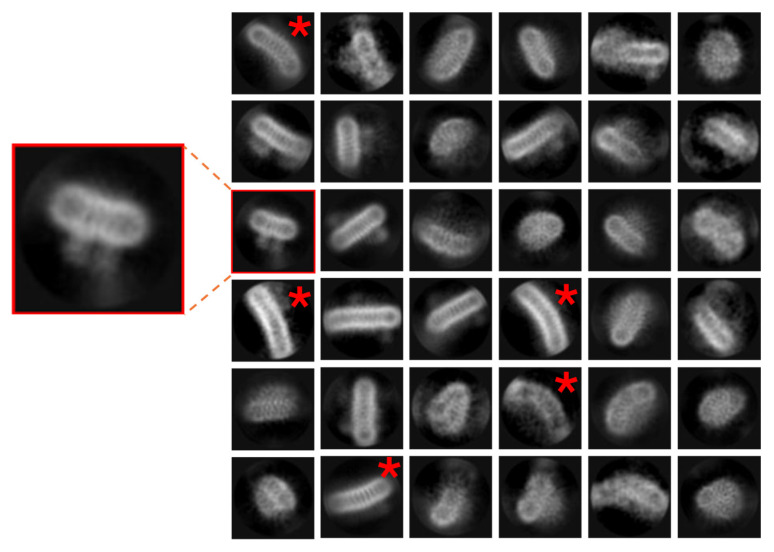
Unusual micellular artefacts and structures. Some detergents (such as 2,2-didecylpropane-1,3-bis-β-D-maltopyranoside (LMNG) used in the example here) are prone to forming an array of micelle structures with different sizes, lengths, and unusual curvature (red *), which can complicate data analysis. Two-dimensional classes for this membrane protein should resemble the projections in the red box. Box size = 170 Å^2^.

**Figure 7 biomolecules-15-01315-f007:**
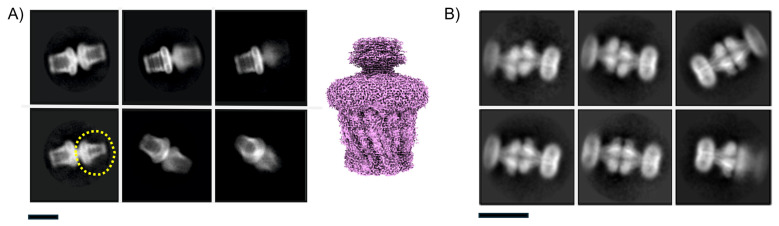
‘Projection contamination’ by non-physiological detergent/buffer-driven interactions may cause processing issues. Pseudo D-symmetrical complexes via membrane-face (**A**) WzzB^ST^ or via cytoplasmic-domain (**B**) McCorA in high-Mg solution. There is also a D form with a mix of oligomeric forms—the smaller oligomer is circled in yellow (bottom-left; the purple surface render shows a 3D reconstruction from particles in (**A**) with no masking to illustrate how the lower layer introduces noise and blur). Scale bars correspond to 200 Å in each panel.

**Figure 8 biomolecules-15-01315-f008:**
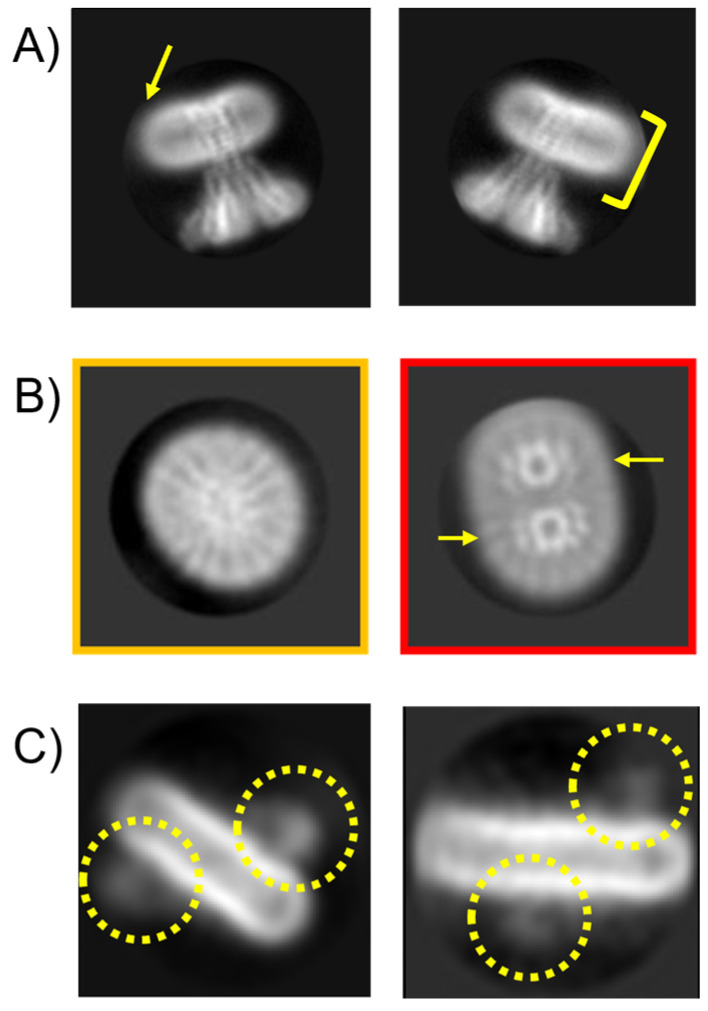
Protein insertion in the micelle can be non-uniform. (**A**) shows 2D classes of McCorA showing asymmetric micelle insertion (yellow arrow) in DDM while the yellow indicator bar indicates some micelle averaging blur caused by variability. (**B**) The 2D class averages of an MFS transporter in DDM micelles. Artefacts appear where in some micelles one protein is present (yellow class) whilst in others two proteins are present (red class—yellow arrows). (**C**) The 2D class averages of a membrane protein in LMNG show up/down insertion into the same micelle: extra-membrane domains are circled in broken yellow. Box sizes correspond to 271, 131, and 170 Å^2^ in (**A**–**C**), respectively.

**Figure 9 biomolecules-15-01315-f009:**
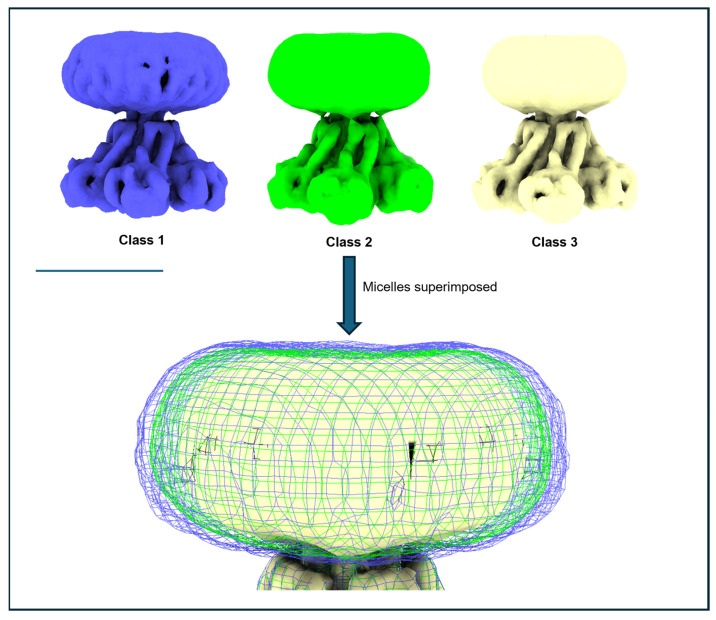
Variable micelle dimensions as a single issue of 3D heterogeneity. In the initial 3D classification of an MjCorA dataset (binned × 4 to ~4.24 Å/pixel), there is very little conformational variability present in the protein; the protein has a cc fit of 0.99 with resolution limited to ~6 Å. At lower resolution sorting, micelle variability is a main source of heterogeneity in 3D samples and the micelle can vary in width by ~20 Å (~15%). The blue and green mesh illustrate this difference in size in superimposition from the corresponding surface rendered volumes above. Scale bar= 100 Å.

**Figure 10 biomolecules-15-01315-f010:**
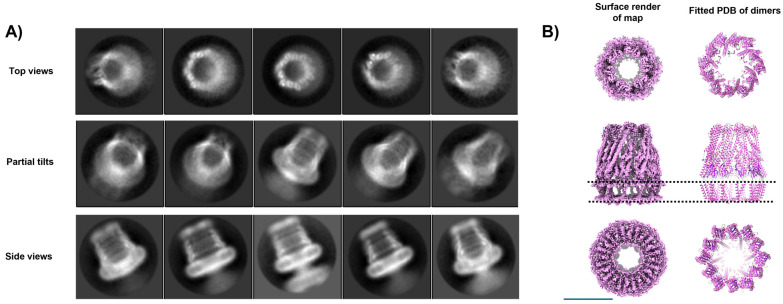
Unusual symmetry effects of a purified Wzz sample. A subset of data from a supplemented dataset of the inner membrane protein Wzz demonstrates a clear C6 sub-arrangement of a dodecamer in (**A**) 2D class averages (box = 200 Å^2^) and (**B**) the resulting 3D map with a fitted atomic model of the Wzz C6 ring (represented as ribbons). The position of transmembrane helices and the inner membrane region is shown by the dotted lines. Scale bar = 100 Å.

**Figure 11 biomolecules-15-01315-f011:**
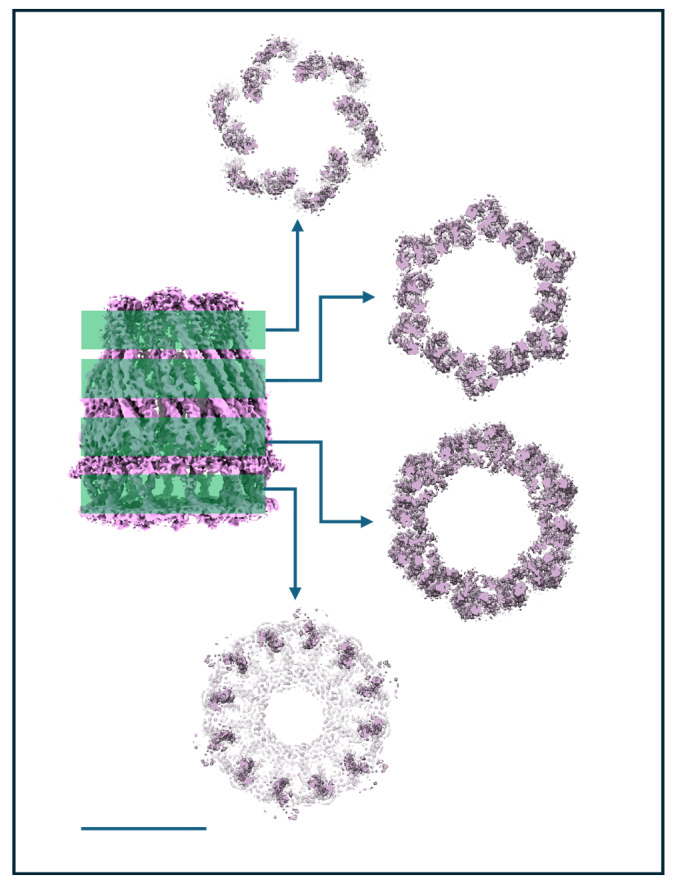
Symmetry strength through the C6 Wzz complex. Slices in different potions of the C6 Wzz structure show the complex has strong C6 deviations in loops and periplasmic helices, while the transmembrane helices within the micelle are only mildly offset from a C12 ring. Scale bar = 100 Å.

## Data Availability

The authors will deposit the Wzz C6 MRC volume and modelling PDB to the EMDB on acceptance. Raw EM data particle stacks will be archived in line with journal policy.

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
