# Peer review of "Coming Clean and Avoiding Bubble Trouble–Using Detergents Wisely in the Purification of Membrane Proteins for Cryo-EM Studies"

_biomolecules, 2025, doi:10.3390/biom15091315_

Round 1
Reviewer 1 Report
Comments and Suggestions for Authors
The article «Coming clean and avoiding bubble trouble – using detergents wisely in the purification of membrane proteins for Cryo-EM studies» by Chen et al. is well-written and informative, and will hopefully help a select target audience (specifically students working in cryo-em of membrane proteins) with troubleshooting their experiments. As stated in more detail below, I miss a more thorough explanation/discussion of the underlying molecular principles for some examples, and I also have some comments to manuscript organization. The biggest “disappointment” to me is that the title of the paper promises some kind of selection guide for detergents, but then only gives examples of things that worked (instead of telling readers what to more systematically test if there is trouble, and explaining the molecular principles behind these changes). I hope that the authors can address this point of criticism and that my comments below are helpful for this.
The paper is classified by the authors as a “commentary research paper” (page 3 line 115). Question for the editor: I am not sure this category exists?
General comments:
- I think the manuscript would very much benefit from an overview figure that shows the different steps of the cry-em sample preparation and data acquisition and data analysis workflow(s), with clear reference in that overview figure what “detergent issues” will appear in what step, potentially.
- I think the manuscript organization could be more based on the above figure. Currently, issues are grouped by the example protein that they are observed with, rather than (only) following the steps of the workflow. See detail comments below.
- I am disappointed that in the section where the different issues are described, this remains on “observation” and “screening/troubleshooting” level. I would have hoped to get some insight or at least some speculation with each ‘issue’ regarding which detergents will cause this issue more, and which ones less, and also what the molecular basis of this may be (often this will be linked to micellar size, CMC, exchange rates between free detergent vs micellar phase, and so on…). If this is meant to be a guide to troubleshooting, such discussion would help the reader to make more informed decisions on what parameter to change?
- Generally speaking, it would be advantageous for each “issue” if its description would always follow the same structure: what is the problem (with the example at hand demonstrating the problem), what is the underlying molecular properties causing it (if known), and guidelines on how to fix or troubleshoot. I really love the examples and they are illustrated extremely well, but it is currently hard to navigate the paper if you look for specific answers to a specific problem.
Detail comments:
P2 line 54: could the authors specify in more detail _why_ these other types of phases (that are not really micelles but stem from other parts of the detergent phase diagrams) are not suitable for single-particle analysis, to make this point more clear?
P2 line 56-70: this section could be a bit more explicit on the direct impact that alkyl chain length has e.g .on CMC sand on “mildness”, as the authors put it.
End of introduction section: I am a bit astonished that the authors do not also discuss saposin-based nanodiscs here (e.g. as an alternative to apolipoprotein disks) (se f.ex. https://www.sciencedirect.com/science/article/pii/S0959440X25000685).
P3 line 82: SMA is a block copolymer, and there are also other block copolymers that have been used for membrane proteins. Maybe introduce the molecule category more generally first?
For “Issue 1” it is problematic (regarding the organization of the text) that the authors summarized several sub-issues in this section (and also call them “issue”… e.g. p8 line 272 “another issue may arise…”). I recommend to treat “Interface accumulation” as a separate issue section (Issue 2? Renumbering later ‘issues’?)?
For “Issue 2” the authors might want to warn the reader that removing excess micelles is only a good idea for detergents with rather stable micelles. Detergents that are easily washed off from detergent-protein complexes (such as beta-OG!) might also do that during excess detergent micelle removal, leaving the membrane protein prone to precipitation! In other words: this may well only work with beta-DDM…
For “Issue 3” it is not entirely clear why this is different from “issue 2” where there is also excess empty micelles. If there is a difference, this should be spelled out more explicitly. If this is the same issue but in different steps of the cry-em workflow,, this could also be made more clear (see general comments above).
Reviewer 2 Report
Comments and Suggestions for Authors
This manuscript focuses on the problems associated with detergents during membrane protein structure determination using the cutting-edge cryoEM single-particle analysis, a very successful method that has significantly improved the structural biology of membrane proteins, a particularly challenging group of biological molecules. This method/protocol-like article or review provided insightful details and valuable strategies for membrane protein sample preparation and single-particle data processing, which is useful. My comments are listed below.
1. Page 1, line 24, "major facilitator super-family" please change to major facilitator superfamily transporters.
2. Page 2, line 62, "Lauryl Maltose Neopentyl Glycol (LMNG) with preferential properties including smaller" citation to the original Nature Methods paper is suggested. Nat Methods 7, 1003–1008, doi: 10.1038/nmeth.1526 (2010).
3. Page 2, line 66, "shown β-DDM". Do you mean β-DDMB?
4. Page 7, line 256, "detergent-purified samples" is specific. The sample can be purified in detergent and then reconstituted into lipidic nanodiscs. Please change to "proteins in detergent solution"
5. The issue of detergent being concentrated during membrane protein preparation due to improper use of concentrators has been well recognized. Detergent concentration in the membrane protein samples is really unknown; using slightly lower than the CMC value or equal to the CMC value has been established.
Reviewer 3 Report
Comments and Suggestions for Authors
I would recommend conducting a more thorough literature review of what has already been published. In structural biology (cryo-em), the most widely used and successful detergents include LMNG + CHS, digitonin, and GDN. Referring to LMNG as a "newer detergent" feels outdated at this point. Additionally, there is no mention of peptidisc or saponin-based approaches (such as DirectMX), and the discussion only references MSP.
Regarding polymers, the manuscript only considers SMA and DIBMA, but there are many others worth noting, such as AASTY, PMA, and NCMNs. These polymers are not widely adopted due to issues like precipitation in the presence of bivalent cations, sensitivity to pH, and relatively low efficiency in extracting membrane proteins from the lipid environment and not because of limited resolution. Furthermore, there are differing viewpoints on whether and when to introduce polymers during purification workflows.
The discussion of UltrAuFoil and Hexafoil grids is also misleading, it implies that both have a hexagonal hole arrangement, which is not the case for UltrAuFoil.
Figure 2 clearly displays camera artifacts; I recommend using a different image for clarity.
While I agree that membrane proteins often show unexpected oligomeric associations, I would avoid stating that they tend to form pseudo-D-symmetrical assemblies, as this is not broadly supported.
Finally, the map and model for Wzz C6 do not fit. The map appears to be flipped and needs to be corrected.
Overall, I believe the manuscript contains several inaccuracies and areas for improvement. It lacks novelty and touches only superficially on the complexities of membrane protein structural biology. For example why not mentioning support films? E.g. Graphene Oxide, Single layer graphene, Continous Carbon and etc.
Comments on the Quality of English LanguageThere are numerous small but noticeable errors throughout the manuscript, for example, inconsistent formatting between citations (e.g., sometimes a space is added before the reference, sometimes not), incorrect use of units such as "Kda" instead of "kDa," and a missing "B" in "β-DDM." The name "Apolipoprotein I ApoA-I" should likely be written as "Apolipoprotein I (ApoA-I)". Two dots in a row. Additionally, the Methods section is poorly written and lacks detail. For example, 3D structures were produced in Relion?. These issues collectively affect the overall quality and readability of the paper.
Reviewer 4 Report
Comments and Suggestions for Authors
The manuscript describes difficulties in cryo-EM studies of membrane proteins arising from the unavoidable use of detergents to extract the protein from the membrane and to keep it in single-particle form in an aqueous solution. It provides helpful information for researchers in the field, in particular for novices that may encounter challenges. In my view, the paper is useful and suitable for publication in biomolecules subject to some minor revision.
In general, the paper should be reworked to ensure that figures appear in the same order they are referred to in the text, and that all information about what is seen is given in the figure captions (e. g. which protein in Figure 1).
Specifically, at the end of line 66 it should probably read ß-DDMB.
In line 121, E. coli should be written as Escherichia coli, and then abbreviated afterwards and always written in italics.
Line 311: What do the authors mean by DDM is disrupted by SDS?
Line 395: Please, explain D-symmetry briefly.
Finally, I have a more special question: In section 3.2, the authors give the impression that it might be possible to separate empty micelles from protein-detergent complexes (PDCs). However, I doubt that this is possible. As correctly stated later in the manuscript, micelles are flexible, dynamic structures. The same is true for the detergent belts surrounding the protein in the PDCs. The key point is that these are non-covalent aggregates that are permanently in equilibrium with detergent monomers in the solution. This is why the CMC must be exceeded to ensure the formation of these structures. Depending on the equilibrium constants of these aggregation processes, there will always be a certain amount of empty micelles in the presence of PDCs. Do the authors have any more quantitative data about how far the concentration of empty micelles can be lowered without jeopardizing the protein solubilization? I guess, quantification of the detergent is difficult. It might be useful to discuss the problem of detergent analytics a bit more, in particular, in the context of using Centricon concentrators.

Round 2
Reviewer 1 Report
Comments and Suggestions for Authors
all of my comments have been addressed at least partially, and I do appreciate why the authors decided to not change the overall structure of hte manuscript significantly (as previously suggested by this referee). Please carefully check for typos in the newly added sections (e.g. it should say "GroEL" in line 440 to my understanding)
Reviewer 3 Report
Comments and Suggestions for Authors
The authors have carefully addressed all of my suggestions in the revised version. The changes improve the clarity and quality of the manuscript. In my view, the paper is now suitable for publication in its current form.